# DACO: Towards Application-Driven and Comprehensive Data Analysis via Code Generation

Xueqing Wu[1]   Rui Zheng[2]   Jingzhen Sha[1]   Te-Lin Wu[1]   Hanyu Zhou[1]   Mohan Tang[1]   Kai-Wei Chang[1]
Nanyun Peng[1]   Haoran Huang[3]

## Abstract

Data analysis is a crucial analytical process to generate in-depth studies and conclusive insights to comprehensively answer a given user query for tabular data. In this work, we aim to propose new resources and benchmarks to inspire future research on this crucial yet challenging and under-explored task. However, collecting data analysis annotations curated by experts can be prohibitively expensive. We propose to automatically generate high-quality answer annotations leveraging the code-generation capabilities of LLMs with a multi-turn prompting technique. We construct the **DACO dataset**, containing (1) 440 databases (of tabular data) collected from real-world scenarios, (2) $\sim 2k$ query-answer pairs that can serve as weak supervision for model training, and (3) a concentrated but high-quality test set with human refined annotations that serves as our main evaluation benchmark. We train a 6B supervised fine-tuning (SFT) model on DACO dataset, and find that the SFT model learns reasonable data analysis capabilities. To further align the models with human preference, we use reinforcement learning to encourage generating analysis perceived by human as *helpful*, and design a set of dense rewards to propagate the sparse human preference reward to intermediate code generation steps. Our DACO-RL algorithm is evaluated by human annotators to produce more helpful answers than SFT model in $57.72\%$ cases, validating the effectiveness of our proposed algorithm.

*Equal contribution   [1]University of California, Los Angeles [2]Fudan University [3]ByteDance AI Lab. Correspondence to: Haoran Huang <huanghaoran@bytedance.com>.

*The first AI for MATH Workshop at the $41^{st}$ International Conference on Machine Learning*, Vienna, Austria. Copyright 2024 by the author(s).

## 1. Introduction

Data analysis is the process of systematically applying statistical and/or logical reasoning to evaluate and comprehend data. Existing literature has investigated answering queries about information given by structural data (*e.g.*, tables) (Chen et al., 2021a; Nan et al., 2022; Lu et al., 2023). However, they either focus on straightforward factual retrieval or short-form entity/arithmetic resolutions for specifically given entities, while real-world data analysis can involve more complex analytical processes.

Take the scenario in Figure 1 as an example: a user is investigating potential age discrimination of a shop. To effectively answer queries such as this one, a chain of mathematical and logical reasoning and interacting with the data is required. For instance, *finding 1* is inferred from analyzing age distribution within the membership data ('member' table), while *finding and suggestion 2* are derived by comparing the participants' ages during the happy hours (using both 'member' and 'happy_hour_member' tables). These rigorous quantitative analyses eventually conclude the opposite to the user's hypothesis. As valuable as the conclusive suggestions such comprehensive analysis can bring, the extensive labor-efforts, hinted by these examples, can hinder the efficiency of gaining intelligence from the data in a competitive business environment. It is thus imperative to devise a system that is able to automate the aforementioned data analysis process.

To this end, we introduce a new dataset for this challenging task, DACO, **d**ata **a**nalysis via **co**de generation. DACO is constructed from a set of diverse real-world databases associated with curated user queries. In light of the previously described labor-intensive challenge, we propose to leverage LLMs with a multi-turn chained prompts to automatically curate the analytical answers for each query. Specifically, our designed framework employs the code generation capabilities of GPT-4 (OpenAI, 2023) for automating the statistical analysis, interleaved with its ability to interpret the obtained quantitative results. The DACO dataset contains $440$ databases and $1,942$ associated user queries, which can be used for both model fine-tuning and evaluation. To provide a refined benchmarking resource, we curate a high-

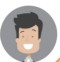
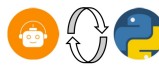
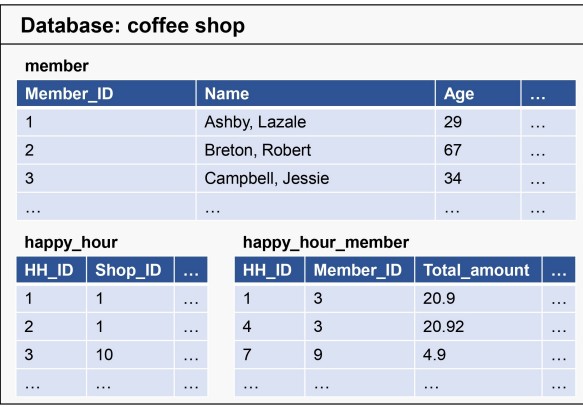

Figure 1: **Task overview.** Given a user query driven by an application scenario, a data analysis system should produce an answer containing findings and suggestions based on the database. This requires the system to perform mathematical, logical and strategic reasoning, which can be done through invoking external tools such as Python libraries. In this example, *finding 1* is inferred from analyzing age distribution within the membership data ('member' table) through mathematical reasoning and analytical reasoning. *Finding 2* is inferred by comparing the ages of the happy hours participants (using 'member' and 'happy_hour_member' tables) through mathematical reasoning, and *suggestion 2* is further derived by relating the data to coffee shop business setting through strategic reasoning.

quality test set through comprehensive human annotations on a subset of 100 samples. Detailed statistics are in Table 1.

Although LLM exhibit reasonable analytical capabilities (and hence we are able to automatically curate the pre-refined data), we empirically find that its generations often fall short of human expectations of what good analyses should be (*e.g.*, relevance to the queries, logical coherency, and higher-level quantitative interpretations). To further improve the generations by aligning the models with corresponding human preferences, we design a reinforcement learning algorithm (DACO-RL) that leverages two newly designed reward models (RM), *i.e.*, *contribution RM* and *regularization RM*, to efficiently provide denser feedback. Concretely, the contribution RM heuristically provides better learning signals for the intermediate code generation steps (for more relevant quantitative analysis), while regularization RM helps preventing reward hacking (Skalse et al., 2022) of typical RLHF models (Casper et al., 2023). We test our algorithm on a fine-tuned CodeGeeX-6B model (Zheng et al., 2023a), where the win rate of 57.72% in human-annotated pairwise comparison justifies its effectiveness on learning to generate human preferred analyses.

In summary, our contributions are three folds: (1) We explore the challenging task of data analysis, where we construct the DACO dataset with our proposed multi-turn prompting technique on a diverse set of real-world databases. (2) We curate a human-refined evaluation set for benchmarking models. (3) We design the DACO-RL algorithm to jointly optimize code generation and answer generation towards human alignment, which demonstrates a significant 57.72%

human evaluated win rate on the helpfulness metric.

## 2. The DACO Task and Dataset

As shown in Figure 1, the input to our task is a database $\mathcal{D}$ and a query $\mathbf{q}$, and the output answer $\mathbf{a}$ is formatted as two lists of findings and suggestions respectively. In this work, the database $\mathcal{D}$ should be a relational database containing multiple named tables.

We construct our DACO dataset through four stages: (1) database collection, (2) query collection, (3) automatic annotation collection, and (4) human refinement. The workflow is illustrated in Figure 2. Our final dataset contains the training, development and test sets with annotations generated by GPT-4, along with a human-refined testing subset. To distinguish the two test sets, we use Test$^A$ to represent the automatically annotated set and Test$^H$ to represent the human refined one. Statistics are shown in Table 1.

**Database collection.** We collect databases from two sources: Spider (Yu et al., 2018) and Kaggle (https://www.kaggle.com/datasets). There are 157 databases collected from Spider, which originally come from university databases, DatabaseAnswers and Wikipedia. We additionally crawl and filter 5,830 databases from Kaggle. From this pool, we manually select a subset of 314 clean and interpretable databases to build our dataset. To maintain the diversity of the resulting database set, 157 of the databases are deliberately chosen near the long tail of its topic distribution. For this, we employ BERTopic (Grooten-dorst, 2022) to model the topic distribution, which produces in total 160 topics. We take its least frequent 80 topics as

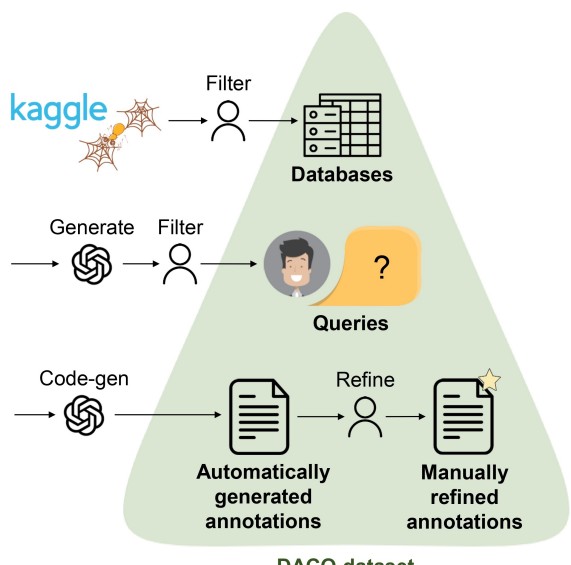

Figure 2: Curation process of DACO dataset.

| | Train | Dev | Test$^A$ | Test$^H$ | Total |
|---|---|---|---|---|---|
| # db | 353 | 22 | 65 | 17 | 440 |
| # queries | 1558 | 100 | 284 | 100 | 1942 |
| **Database size** | | | Med. | Max | Min |
| # tables | | | 1 | 15 | 1 |
| # columns | | | 6 | 50 | 3 |
| # rows | | | 20 | 19,237 | 2 |
| **Answer size in Test$^H$** | | | Med. | Max | Min |
| # findings | | | 5 | 8 | 3 |
| # suggestions | | | 5 | 8 | 3 |
| # tokens | | | 397 | 864 | 202 |

Table 1: **Statistics of DACO dataset.** Train, Dev and Test$^A$ sets are automatically generated with GPT-4, while Test$^H$ is the human refined subset. We report the size of each data split, the size of input databases, and the size of output answers in human refined test set (Test$^H$).

the long tail, which covers 26.79% of the total databases.

In total, DACO comprises 471 databases, each of which contains on average 2.3 tables. To better visualize the *major topic* distribution of this selected subset, we again use BERTopic but group these databases into 10 topics. The keywords for top 5 topics are shown in Figure 3. The leading topic (topic 1) is associated with business setting and consists of 46.52% of the dataset. The remaining nine topics exhibit a relatively even distribution, covering a broad range of domains, including sports (topic 2), healthcare (topic 3), weather (topic 4), and education (topic 5).

**Query collection.** We generate 10 queries for each database by prompting ChatGPT to first assume the role of a database stakeholder and then generate an application-driven query based on the role. To ensure the quality of the query, we perform a manual filtering to the machine generated queries. Specifically, we remove queries that are not driven by real-world applications or cannot be answered by the given reference database. We train a group of 6 annotators to perform such a filtering process. As a result, there are about 42% of the queries removed, where the removal agreement achieves a 0.62 cohen kappa score.

After the aforementioned processes, we obtain in total 2,664 queries. We show the top 15 verbs and their top 3 direct noun objectives in Figure 2. The queries demonstrate a notable level of diversity. The most common type of queries is to request analysis (such as "analyze data" and "identify pattern"), followed by queries aiming to make decisions (such as "determine strategy" and "make decision").

**Automatic annotation collection.** As shown in the right half of Figure 4, we design a pipeline that leverages the code generation capability of LLMs to automate the answer

annotation for our DACO dataset. Based on the database and the query, we instruct the LLM to perform data analysis in multiple turns. At each turn, the LLM will produce a python code snippet and take its execution outputs as evidences to reason over and support its follow-up interpretation. After each turn, we prompt the model to decide whether the analysis is sufficiently comprehensive; if deemed sufficient, it terminates the coding turns and produces the final answer.

With this pipeline, we instruct GPT-4 to automatically generate all the answer annotations to each query of our dataset, for both the intermediate code and the final analysis answering the queries. To improve the quality of such automatically constructed annotations, we additionally allow GPT-4 to correct its own mistakes when its generated code leads to run-time or syntax error, where only the corrected codes are kept. In total, we obtain 1.9k valid query-answer pairs, each with roughly 3.3 intermediate coding steps.

**Human refinement.** The annotated analyses thus far have been algorithmically generated, where their actual quality are to be further verified. We thus curate a human-refined subset containing 100 densely human-annotated query-answer pairs. For each query, we sample 3 different analysis candidates using the previously described automated method (with GPT-4). We ask the annotators to evaluate the quality of each machine generated bullet point and categorize each point into one of bad, borderline, or good (associated with scalar scores of 0, 1 and 2 respectively, the higher the better) to each. The bullet points deemed higher quality are then mixed (from the 3 candidate analyses) and refined (with a few manual textual edits) into one final gold-analysis. In the refinement stage, the annotators should first combine all bullet points ranked as "good", remove duplicate points, and reorder the points to maintain a coherent flow. Suppose the number of bullet points are lower than our pre-defined lowest threshold (3 bullet points per answer), the annotators should select bullet

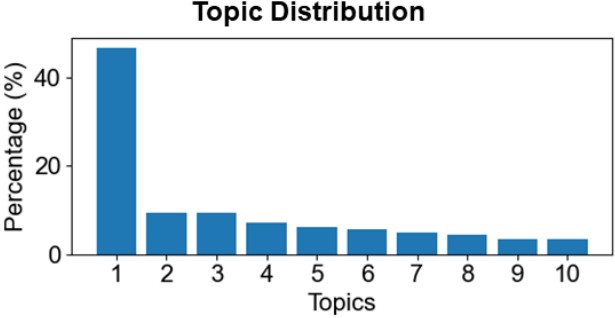

**Keywords for Top 5 Topics**

Topic 1: price, sales, customer, company

Topic 2: player, game, club, winners

Topic 3: world, health, life, expectancy

Topic 4: weather, temperature, rainfall, forecasting

Topic 5: student, school, universities, education

Figure 3: **Domain distribution of DACO databases.** We display the topic distribution and keywords for the leading 5 topics. Topics are extracted from database titles using BERTopic. This demonstrates the diverse domain coverage of DACO.

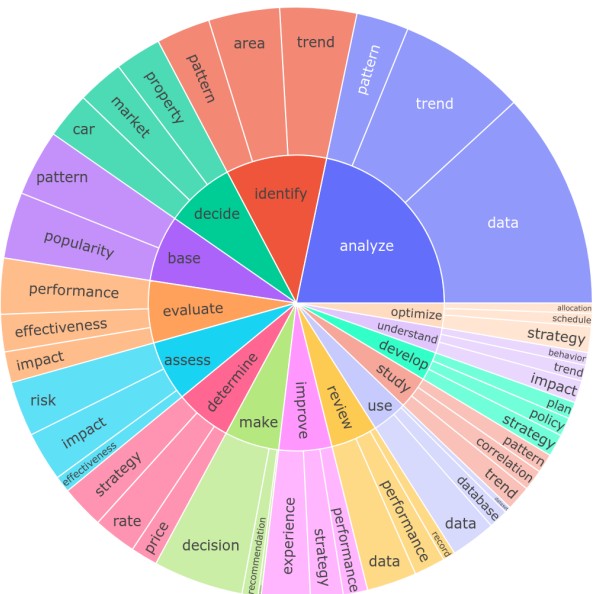

Table 2: **Distribution of DACO queries.** We display the top 15 verbs and their top 3 direct noun objectives, demonstrating the diversity of DACO queries.

points ranked as "borderline" to augment the answer. we ask a group of 3 internal members to perform refinement. The agreement accuracy of the refinement process (candidate point selection) is 0.83 and the Cohen's Kappa is 0.67.

**Evaluation.** To evaluate the quality of generated data analysis, we use **helpfulness** as the main metric. Motivated by literature in the data analysis field (Long & Long, 2009), we define helpfulness as: (1) relevance to the query, (2) effective and insightful data interpretation, and (3) diversity in terms of analysis perspectives. We evaluate helpfulness through **pairwise comparison** following common approach (Ouyang et al., 2022; Wu et al., 2023; Zheng et al., 2023b). Given two analyses generated by two different systems, the annotator (either human or simulated by ChatGPT) selects the more helpful one based on our defined criteria. The winning rate of each system is reported as helpfulness score. To obtain a comparable set of numbers for all models, we report the winning rate of each model against Test[A] and Test[H] annotations. The upper bound for this score would be 50, as a score of 50 indicates that the model generations are perceived as helpful as annotations.

## 3. DACO-RL

While DACO contains mostly algorithmic machine generated analyses, the machine generations without human refinement cannot well align with human preferences (of "good" analyses). Our human refinement process shows that only 47.4% bullet points are "good" points perfectly addressing user queries; the majority of 52.2% are evaluated as "borderline" points that only partially aligns with

human expectations; and the remaining 0.4% are considered as "bad".

We are therefore interested in investigating whether aligning human preferences via an RLHF fashion could lead to better machine generated analyses. We thus propose the DACO-RL algorithm, which is illustrated in the left half of Figure 4. Our end goal is to optimize the helpfulness of the analyzed points, which is modelled with an *answer RM* $R_a$. In addition to this sparse reward signal, we use a heuristically defined *contribution RM* $R_c$ to reward each intermediate step, which is further regularized with a *regularization RM* $R_r$ to prevent reward hacking. In the following sections, we first explain the three reward models sequentially, and eventually explain our whole RLHF pipeline.

**Notations.** We train a language model that interacts with the python interpreter in a conversational manner. Formally, the full dialogue is a list of messages $[\mathbf{h}, \mathbf{c}_1, \mathbf{o}_1, \cdots, \mathbf{c}_m, \mathbf{o}_m, \mathbf{a}]$, where $\mathbf{h}, \mathbf{c}_i, \mathbf{o}_i$ and $\mathbf{a}$ stand for human message, code, execution outputs and final answer respectively. The dialogue starts with human message $\mathbf{h}$ containing both query $\mathbf{q}$ and database meta-data. In the later messages, code $\mathbf{c}_i$ and final answer $\mathbf{a}$ are generated by our model, while execution outputs $\mathbf{o}_i$ are produced by the python interpreter. To feed the dialogue into the language model, we wrap each message between a begin-of-message phrase (denoted as <BOM>) and the end-of-sentence token, and then concatenate all messages together. We use a different <BOM> for each type of message. Thus, at each turn of the conversation, the language model can decide whether to generate code $\mathbf{c}$ or final answer $\mathbf{a}$ by generating

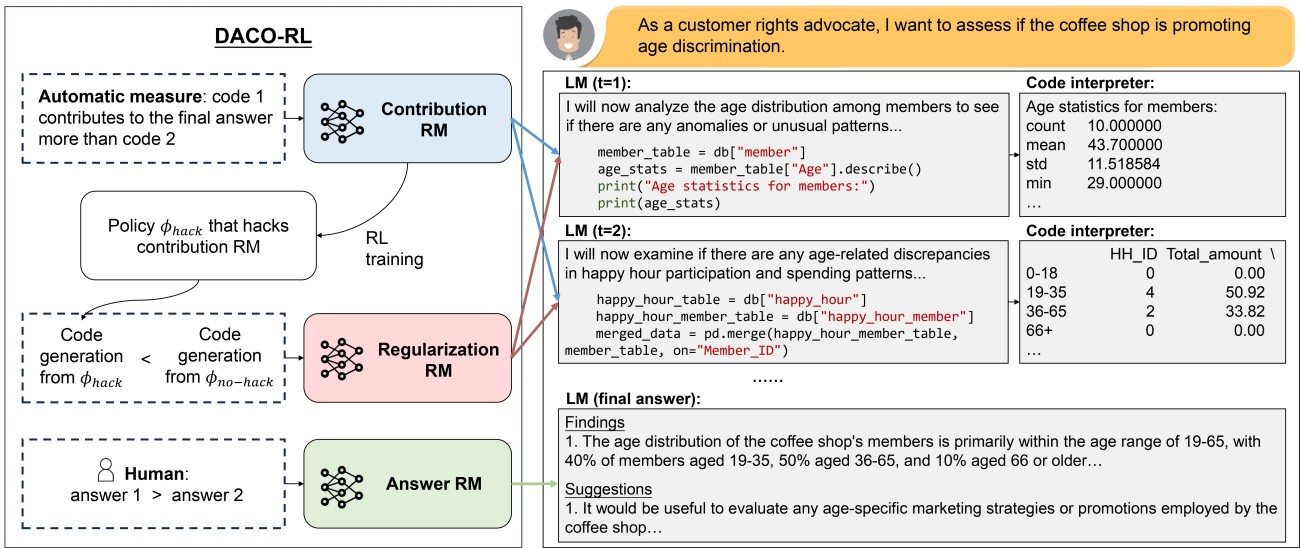

Figure 4: **Overview of code generation pipeline (right) and our DACO-RL algorithm (left).**
**Code generation (right)**: At each turn, the model generates python code, execute the code with a code interpreter, and reads the execution outputs. Eventually, it summarizes the results into a final answer.
**DACO-RL (left)**: The answer RM evaluates the helpfulness of the final answer and is our end optimization goal. However, to provide denser reward, we use a contribution RM to evaluate how much each generated code contribute to the final answer. The reward is provided at the end of each generated code snippet. Contribution RM is vulnerable to reward hacking, which motivates us to propose regularization RM to discourage reward hacking generations.

the corresponding <BOM>.

**Answer RM.** Our end goal is to optimize the helpfulness of answer $\mathbf{a}$, which is modelled by the answer RM $R_a$. Particularly, we model the helpfulness of each single bullet point rather than the full answer. We collect pairwise comparison data of bullet points from ChatGPT to train $R_a$. Given a pair of bullet points where one is preferred over the other, $R_a$ is trained to assign a higher score to the preferred bullet point. Given access to the full conversation, $R_a$ produces a reward score at the end of each bullet point in the answer.

To encourage diversity, we additionally add a repetition penalty. Given a list of findings $[\mathbf{f}_1, \cdots, \mathbf{f}_F]$, we encourage the $i$-th finding to be different from all previous findings by imposing a penalty score as $\sum_{j=1}^{i-1} Sim(\mathbf{f}_i, \mathbf{f}_j)$, where $Sim(\cdot, \cdot)$ computes the similarity between two bullet points.[1] This score is then subtracted from the reward score from $R_a$. The same procedure is applied to the suggestion list.

**Contribution RM.** The answer RM described above is a sparse reward signal that rewards only the answer $\mathbf{a}$ but not intermediate coding steps $\mathbf{c}_i$. To provide denser reward signal for optimizing $\mathbf{c}_i$, we aim to evaluate the helpfulness of each coding step $\mathbf{c}_i$. However, annotating helpfulness of intermediate steps is much more difficult. For one, the

---
[1] $Sim(\cdot, \cdot)$ is computed as the cosine similarity of Sentence-BERT (Reimers & Gurevych, 2019) embeddings. We use the ALL-MINILM-L6-V2 model.

helpfulness of python code is more vague to define, and the other, evaluating helpfulness requires coding expertise, which makes it more expensive to collect human annotations.

To measure the helpfulness of intermediate steps without the huge expense required by human annotations, we heuristically define the helpfulness as **how much an intermediate step contributes to the final answer**. Concretely, we compute the similarity $Sim(\mathbf{a}, \mathbf{o}_i)$ between final answer and code outputs to measure the helpfulness of $\mathbf{c}_i$. We use $Sim(\mathbf{a}, \mathbf{o}_i)$ to rank the helpfulness of different steps, and use the comparison pairs between intermediate steps to train the contribution RM $R_c$. Given the conversation, $R_c$ predicts the contribution level of each $\mathbf{c}_i$ as its reward score. $R_c$ does not take the execution output $\mathbf{o}_i$ into consideration when scoring $\mathbf{c}_i$, which simplifies model implementation and excludes spurious correlation the model may exploit from the surface form of $\mathbf{o}_i$.

**Regularization RM.** The heuristically defined contribution RM $R_c$ may not necessarily perfectly align with the true helpfulness. This is due to a known reward misspecification issue termed reward hacking (Skalse et al., 2022), where the policy model achieves higher scores from the reward model but its true reward decreases. We propose to regularize such behavior with a regularization RM $R_r$. Given the misspecified reward model, $R_c$ in our case, we first train an RL model until its generations start to collapse to certain patterns. These patterns typically receive high rewards from

$R_c$ but do not align well with human expectation, and thus are considered as reward hacking behaviors. We denoted this RL model as $\pi_{\text{hack}}$. We use $\pi_{\text{hack}}$ to produce generations with typical reward hacking behaviors. These generations are paired with generations without reward hacking behaviors, such as generations from supervised fine-tuning (SFT) model or the pre-human refined answers generated from GPT-4, to further train the regularization RM $R_r$. As an intuition, this means $R_r$ will assign lower scores to typical reward hacking behaviors.

**RLHF.** For our whole DACO-RL pipeline, we optimize the language model against the mixture of all three aforementioned rewards, $R_a$, $R_c$ and $R_r$.

More specifically, we first train a multi-task reward model denoted as $R_{a+c}$ to jointly learn $R_a$ and $R_c$. This guarantees that the reward score distribution for $R_a$ and $R_c$ are relatively close, so the reward signal for $\mathbf{a}$ and $\mathbf{c}_i$ will not overshadow each other. We train a separate model to learn $R_r$, and mix $R_{c+a}$ and $R_r$ into $\frac{1}{2}(R_{c+a} + w_r R_r)$ when rewarding $\mathbf{c}_i$.[2] Here, $w_r$ is a hyper-parameter to balance the variance of $R_{c+a}$ and $R_r$ tuned to maximize reward model accuracy on development set. This guarantees the mixed reward can both encourage high contribution and penalize reward hacking.

We use proximal policy optimization (PPO) (Schulman et al., 2017) as our learning algorithm. During training, PPO jointly optimizes a value model $V(s)$ and a policy model $\pi(s)$. The objective of the policy model is to optimize the generalized advantage estimation (Schulman et al., 2016) $\hat{A}_t = \sum_{l=0}^{\infty}(\gamma\lambda)^l \delta_{t+l}$, where $\delta_t = r_t + \gamma V(s_{t+1}) - V(s_t)$ for each time step $t$. When applied to text generation, the generative language model is the policy model $\pi$ and each generated token is an action. In our multi-turn conversational setting, however, only part of the tokens in the dialogue are generated by language model (concretely, $\mathbf{c}_i$ and $\mathbf{a}$). In other words, although the language model still takes the full conversation as input, we only compute GAE and gradients over the model generated subsequence, *i.e.*, $[\mathbf{c}_1, \cdots, \mathbf{c}_m, \mathbf{a}]$.

# 4. Experiments

The goal of our experiments is to verify that (1) augmenting language model with code generation can benefit data analysis, and (2) DACO-RL can further boost the answer helpfulness. To this purpose, we perform the following experiments:

**Evaluated systems.** We evaluate the code generation pipeline with ChatGPT and GPT-4. With the answer annota-

---

[2]In practice, we find $R_r$ has a large non-zero mean value, so we subtract it before weighted average.

tion generated by GPT-4, we further train a 6B CodeGeeX2-6B (Zheng et al., 2023a) model through both SFT and DACO-RL. For each of these models, we experiment with a baseline counterpart that does not include code generation and instead directly takes raw table content as input. We additionally experiment with two models specifically pre-trained on tabular data, TAPAS (Herzig et al., 2020) and TAPEX (Liu et al., 2022). TAPAS is a BERT-style model pre-trained to select relevant information from a table based on user query. For our dataset, we first use TAPAS to select relevant information and then use ChatGPT to interpret the selected information. TAPEX is a pre-trained encoder-to-decoder model. We fine-tune TAPEX with GPT-4-generated annotations.

**Evaluation.** The main metric we use is **pairwise comparison of helpfulness** as in Section 2. We use both ChatGPT and trained human annotators for the evaluation. We additionally report BLEU score, entailment score, and helpfulness evaluation for each individual bullet point. These metrics cannot holistically measure the analysis helpfulness, but can provide complementary insights for analyzing model performance. For entailment, we use an off-the-shelf NLI model to compute the probability that the model generation is entailed by the annotation. For point-wise evaluation, we ask the annotator to assign a score chosen from 0, 1 and 2 to each bullet point using the same standard as in human refinement of test set. Our human annotation achieves high agreement of 0.62 Cohen's kappa for pairwise comparison of helpfulness, and 0.65 Cohen's kappa for point-wise helpfulness evaluation.

## 4.1. Results

The main results are in Table 3. We have the following observations:

**Code generation significantly helps data analysis, especially for zero-shot LLMs.** ChatGPT and GPT-4 both enjoy a significant gain in most metrics, especially helpfulness, when equipped with code generation. As in Table 5, human evaluation further shows that GPT-4 with code generation has a significant 66.41 win rate over GPT-4 w/o code generation. After SFT, code generation brings less significant improvements, because SFT w/o code generation can simulate the behavior in GPT annotations and achieve competitive helpfulness. However, its mathematical and logical reasoning are not supported by code generation, so it produces more hallucination as reflected by the low entailment score on Test$^H$.

**Our SFT model learns reasonable data analysis capabilities.** By simulating GPT-4 behaviors, SFT with code generation achieves a reasonable helpfulness score and outperforms the TAPEX baseline, but still falls short compared to ChatGPT. We also evaluate the error rate (%) of generated

| | Method | # para. | Code gen | Test$^A$ | | | Test$^H$ | | |
|---|---|---|---|---|---|---|---|---|---|
| | | | | Help. | Entail. | BLEU | Help. | Entail. | BLEU |
| *TableQA* | TAPAS | 337M | ✗ | **25.00** | 1.96 | 11.62 | **24.50** | 3.67 | 9.73 |
| *Baselines* | TAPEX | 406M | ✗ | 14.79 | **3.34** | **14.60** | 6.00 | 3.50 | **13.81** |
| *Prompt-* | ChatGPT | 20B$^\dagger$ | ✗ | 25.18 | 3.06 | 13.22 | 18.50 | 2.07 | 13.51 |
| *based LLM* | GPT-4 | 175B$^\dagger$ | ✗ | 30.81 | 3.35 | 14.90 | 24.00 | **4.36** | 13.71 |
| | ChatGPT | 20B$^\dagger$ | ✓ | 35.74 | 2.74 | 14.22 | 27.27 | 2.59 | 14.51 |
| | GPT-4 | 175B$^\dagger$ | ✓ | **52.00** | **4.59** | **17.77** | **41.88** | 3.26 | **17.54** |
| *Finetuned* | SFT | 6B | ✗ | 21.51 | 2.30 | 14.47 | 9.50 | 2.65 | 13.63 |
| *LLM* | SFT | 6B | ✓ | 20.95 | 2.15 | **14.88** | 11.54 | 4.47 | **14.60** |
| | DACO-RL | 6B | ✓ | **28.54** | **3.65** | 13.13 | **21.05** | **5.98** | 11.80 |

Table 3: **Main results.** We report helpfulness (Help.), entailment (Entail.), and BLEU on both automatically annotated test set (Test$^A$) and human curated test set (Test$^H$). We also report the number of parameters (# para.) of each model. †: For ChatGPT and GPT-4, we report the number of parameters based on our best estimation.

| | Answer | | | | Code | | | |
|---|---|---|---|---|---|---|---|---|
| | Help. | Help.-SFT | Entail. | BLEU | Info.-SFT | # lines | # API | Error |
| SFT | 11.54 | 50.00 | 4.47 | **14.60** | 50.00 | **41.17** | 134 | 3.08 |
| DACO-RL | **21.05** | **58.49** | **5.98** | 11.80 | **57.86** | 39.66 | **145** | **2.75** |
| w/o $R_r$ | 18.75 | 51.92 | 3.21 | 13.17 | 52.56 | 32.54 | 120 | 3.84 |
| w/o $R_c, R_r$ | 8.79 | 40.00 | 3.13 | 11.46 | 44.57 | 14.30 | 108 | 3.91 |

Table 4: **Ablation study** of regularization RM ($R_r$) and contribution RM ($R_c$) in DACO-RL. We report helpfulness (Help.), entailment (Entail.) and BLEU scores evaluated on Test$^H$. We also compare the helpfulness of each model directly against SFT and report the win rate (Help.-SFT). For evaluating code generation, we report the informativeness win rate over SFT generations (Info.-SFT), number of code lines (# lines), number of different API (# API), and code error rate per step. Pair-wise comparison results are all obtained from ChatGPT.

| | Pairwise comparison | | Point- |
|---|---|---|---|
| | Human | ChatGPT | wise |
| GPT-4 code gen *v.s.* | **66.41** | **70.07** | **1.45** |
| GPT-4 w/o code gen | 33.59 | 29.93 | 1.36 |
| DACO-RL *v.s.* | **57.72** | **58.49** | **1.42** |
| SFT | 42.28 | 41.51 | 1.30 |

Table 5: **Human evaluation.** We report human-rated and ChatGPT-rated helpfulness pairwise comparison of two pairs of models: GPT-4 with v.s. without code generation, and DACO-RL v.s. SFT. We also report point-wise evaluation scores scaled into 0 ∼ 2 rated by human annotators.

| Top 4 APIs | | Bottom 4 APIs | |
|---|---|---|---|
| API | Corr. | API | Corr. |
| print | 44.24 | to_datetime | -18.96 |
| nlargest | 20.06 | isnull | -17.76 |
| mean | 14.56 | describe | -12.02 |
| sort_values | 12.23 | merge | -10.83 |

Table 6: **APIs** ranked by its correlation with contribution RM scores. Higher correlation means that contribution RM assigns higher scores to code snippets containing the API.

code per step. Our SFT model has an error rate of 3.08%, which is reasonable low, but still much higher than ChatGPT (0.495%) and GPT-4 (0.491%).

**DACO-RL significantly improves over SFT.** As shown in Table 3 and 4, DACO-RL significantly boosts the performance. Despite the difference in model size, DACO-RL outperforms ChatGPT w/o code generation on helpfulness and entailment metrics. When matched in size, DACO-RL significantly outperforms SFT by 7 points on helpfulness, further demonstrating its benefits. Human evaluation demonstrates a 57.72 win rate of DACO-RL over SFT as in Table 5. Our qualitative analysis shows that DACO-RL better focuses on user query, while SFT tends to display generic statistics that are less relevant to user query. An example is shown in Figure 5.

We further perform ablation study and report the results in Table 4. To directly compare the ablation models against SFT, we report the win rate of each model over SFT model.

To evaluate the quality of code generation, we use ChatGPT to compare the "informativeness" of generated code against SFT, where informativeness refers to producing informative and insightful code execution outputs while staying relevant to the user query. We report a few additional statistics including the number of code lines and the number of different API calls. We observe that without our proposed two reward models (DACO-RL w/o $R_c, R_r$), using only answer RM significantly hurts the model generation, leading to short and less diverse code generation and thus less helpful final answers. Contribution RM and regularization RM encourage more diverse code generation and more helpful final answer production.

**Contribution RM favors API calls that extracts important information from tabular data but is also vulnerable to reward hacking.** We report the Pearson correlation between API occurrence and contribution RM scores in Table 6. The functions rewarded most are related to extracting significant features (`nlargest`, `sort_values`), aggregating results (`mean`), and displaying specific information (`print`). In contrast, the least rewarded functions involve

displaying generic statistics (`describe`) and wrangling data (`merge`, `to_datetime`, `is_null`) since they cannot directly contribute to the user query. Examples of generated code and their contribution RM scores are shown in Figure 6a and 6b. However, we notice the concerningly high correlation between `print` function and contribution RM scores, which indicates the RL policy may exploit the correlation to hack reward. Figure 6c shows a typical reward hacking case, where the model achieves a high contribution RM score by printing. Our regularization RM learns to discourage such behavior and helps fix the gap.

**Evaluation on external test sets.** To further analyze the effectiveness of DACO-RL, we evaluate our SFT and DACO-RL models on two external test sets: (1) data analysis benchmark InfiAgent-DA (Hu et al., 2024) (a concurrent work of ours), and (2) free-form table question answering dataset FeTaQA (Nan et al., 2022). We find that DACO-RL improves the accuracy over SFT on InfiAgent-DA (14.61 v.s. 12.92), especially over questions about summary statistics (14.86 v.s. 10.80) and correlation analysis (21.57 v.s. 14.86), which aligns with our evaluation results on DACO-RL dataset. On FeTaQA, DACO-RL retains similar performance (6.35 Rouge-L, 80.74 BERTScore) compared to SFT (6.39 Rouge-L, 80.68 BERTScore) since DACO-RL is not specifically trained to enhance information lookup capabilities.

## 5. Related Work

**Table Analysis.** Early work in table question answering (table QA) targets simple questions that requires table lookup and cell aggregations (Pasupat & Liang, 2015; Zhong et al., 2017; Iyyer et al., 2017; Yu et al., 2018; Nan et al., 2022). Later benchmarks further require free-form answer generation (Nan et al., 2022), multi-hop reasoning (Chen et al., 2021a; 2020) and mathematical reasoning (Zhu et al., 2021; Chen et al., 2021b; Lu et al., 2023). Despite the similar formulation between our task and existing table QA work, their focus are different: most existing table QA datasets focus on obtaining specific information, our data analysis queries can be complex and requires query decomposition and reasoning. Some concurrent work further targets comprehensive table analysis such as correlation analysis and causal reasoning (Nan et al., 2023; Hu et al., 2024; Liu et al., 2024). The main difference between this work to the concurrent work is our focus on addressing application-driven user queries.

**Code Generation.** Code generation benchmarks have been proposed for general-purpose programming (Austin et al., 2021; Hendrycks et al., 2021), math problems (Austin et al., 2021), and data science scenario (Lai et al., 2022; Huang et al., 2022). Similar to our work, some recent work allows the language model to interact with a code execution environment and receive execution outputs as feedback (Yang

et al., 2023; Wang et al., 2023). The most relevant work is Cheng et al. (2023) that also addresses data analysis via code generation. Given a data analysis query, they use GPT-4 to first generate code and then provide an interpretation of the execution results. While their analysis queries are still relatively simple, this is an early exploration aiming at automating data analysis.

**RLHF.** Reinforcement learning from human feedback (RLHF) aims to optimize a language model against human preference (Ouyang et al., 2022; Touvron et al., 2023; Bai et al., 2022a;b; Ziegler et al., 2019; Wu et al., 2023). While traditionally RLHF uses a holistic reward score for the entire generation (Ziegler et al., 2019; Ouyang et al., 2022), recent work shows that dense reward scores for the intermediate reasoning steps are better learning signals (Lightman et al., 2023; Wu et al., 2023). These work uses expensive human annotation to collect annotations for the dense reward data. Compared to human preference, heuristic rewards are more accessible but may not align well with true reward. This gap can lead to reward hacking (Skalse et al., 2022; Pan et al., 2022). A common remedy is to use manually designed heuristics to penalize behaviors that potentially harm the true reward (Ouyang et al., 2022; Laidlaw et al., 2023). In this work, we train the regularizatoin RM to discourage reward hacking.

## 6. Conclusion

In this work, we propose a novel and challenging data analysis task, which involves decomposing user query into multiple perspectives, grounding each perspective to the input data and performing logical and mathematical reasoning. To support this task, we build the DACO dataset containing large-scale annotations automatically generated by GPT-4 and a small but high-quality test set with human curated annotations. We employ LLM enhanced with code generation to this task and evaluate three models on our dataset: zero-shot ChatGPT, zero-shot GPT-4 and a 6B SFT model. While GPT-4 consistently performs the best, SFT achieves reasonably good helpfulness with much less computation. On top of the SFT model, we further proposed our DACO-RL algorithm that significantly boosts the human evaluated helpfulness.

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

# Appendix

# A. Additional Analysis of DACO Dataset

We perform additional analysis to verify the quality of our DACO dataset. We assess data quality based on comprehensiveness and agreement among annotators, which are two of the most commonly considered factors (Ehrlinger & Wöß, 2022).

We first measure the **overlap between input queries** to verify the diversity of automatically generated input queries. We compute the overlap among multiple queries over the same database using cosine similarity of Sentence-BERT embeddings. We use the ALL-MINILM-L6-V2 model. We

find that 46% pairs of generated queries have large difference. A small portion (2%) are repetitive with high similarity; since the percentage is small, it should not seriously affect the dataset quality. Details and qualitative examples are shown in Table 8.

|                        | SFT  | RL   |
|------------------------|------|------|
| learning rate          | 1e-5 | 2e-6 |
| gradient accumulation  | 4    | 4    |
| total steps            | 600  | 200  |
| $\lambda$              | -    | 1.0  |
| $\gamma$               | -    | 1.0  |

Table 7: **Hyperparameters.**

We further evaluate the **comprehensiveness of input queries**, i.e. how many data columns are covered by the analysis. We apply heuristic rules to measure . On average, each analysis covers 71% data columns in the corresponding database. Among all data columns, 90% are covered by at least one data point. This verifies that our dataset achieves good coverage of the database columns.

Regarding **agreement among annotators**, as mentioned in the main content, the machine-generated queries are filtered by human annotators with a 0.62 Cohen's kappa score, and our manual refinement of the test set also achieves a substantial 0.67 Cohen's kappa score. These also verify the quality of our DACO dataset.

## B. Implementation Details

For **zero-shot API-based systems** including ChatGPT and GPT-4, we evaluate two settings, directly reading the table content, and using code generation. For the former setting, we linearize the table content into text representation as model input. Due to token limit, we feed the first 20 rows as input, which covers the full content of 93% tables. For the code generation setting, we employ the pipeline described in Figure 4. When the generated code causes a syntax or runtime error, we re-sample the model until the generated code can be executed. We allow up to 5 resamplings for each turn. We use the `gpt-3.5-turbo-16k-0613` API for ChatGPT and `gpt-4-32k` API for GPT-4. We limit the number of total coding turns maximally at 9. For annotation generation where GPT-4 self-correction is allowed, we limit the number of self-correction within 2 for each turn and 4 for the whole session.

For **finetuned models** including DACO-RL and SFT, we use CodeGeeX2-6B (Zheng et al., 2023a) as the base model. We first train the SFT model using GPT-4 annotations, and then train our DACO-RL model on top of the SFT model. When training $R_{a+c}$ and $R_r$, we initialize the model from the SFT model. When training our DACO-RL model, we initialize

the value model $V$ from $R_{a+c}$, and initialize the policy model $\pi$ from the SFT model. In inference, we use nucleus decoding with p = 0.9 and temperature = 1.0. Similarly, we allow up to 5 resamplings when the generated code causes an error. The SFT model is trained with 8 A100 GPU for about 4 hours. The DACO-RL model is trained with 8 A100 GPU for about 18 hours. Detailed hyper-parameters are in Table 7. The only hyper-parameter we tune is $\lambda$ for DACO-RL. We experiment with 0.8, 0.9 and 1.0 and discover that 1.0 works the best.

For **experiments on external test sets**, we directly evaluate the trained SFT and DACO-RL model on InfiAgent-DA (Hu et al., 2024) and FeTaQA (Nan et al., 2022) test sets. For InfiAgent-DA, following the original paper, we add a reformatting step to reformat the generated data analysis report into the key-value format. We use ChatGPT to perform reformatting with a simplified prompt without in-context examples.

## C. Qualitative Examples

We show final answers generated by SFT and DACO-RL in Figure 5. DACO-RL better focuses on user query, while SFT tends to display generic statistics that are less relevant to user query.

We show examples of code generations in Figure 6. We also report their reward scores from contribution RM and regularization RM.

## D. GPT Prompts

Here we show the prompts we use for ChatGPT and GPT-4. Prompt for query generation is in Table 9. Prompt for helpfulness annotation collection is Table 10. Prompts for helpfulness and informativeness evaluation are Table 11 and 12.

| | % | Example |
|---|---|---|
| Large difference (similarity<0.5) | 46 | As a weather forecaster, I want to study the correlation between weather conditions and bike rentals.  *v.s.* 
 As a tourist attraction planner, I want to evaluate the bike-sharing program's impact on tourism and visitor satisfaction. 
 *Similarity = 0.42* |
| Medium difference (0.5<similarity<0.8) | 52 | As a farmer, I want to determine the suitable fruit varieties to grow on my farm.   *v.s.* 
 As a fruit exporter, I want to identify the fruits that meet export standards and have a longer shelf life. 
 *Similarity = 0.69* |
| Repetitive (similarity>0.8) | 2 | As a consultant for honey market, I want to study the honey production trend to recommend business strategies for my clients.   *v.s.* 
 As a curious analyst, I want to study the production trend to understand the US honey industry. 
 *Similarity = 0.85* |

Table 8: Cosine similarity and qualitative examples of pairs of input queries.

**As a game developer, I want to balance the stats of different Pokemon to ensure fair gameplay.**

| SFT | DACO-RL |
|---|---|
| **Findings** 
 1. There are 8 attributes (height, weight, hp, attack, defense, sp atk, sp def, speed) in the dataset that have strong positive or negative correlations, indicating that certain attributes are related. 
 2. The pairs of attributes with strong positive or negative correlations are as follows: 
     - height and weight (correlation: 0.65) 
     - hp and attack (correlation: 0.45) 
     - hp and defense (correlation: 0.44) 
     - hp and speed (correlation: 0.34) 
 3. The pairs of attributes with strong negative correlation include: 
     - sp atk and defense (correlation: -0.6) 
     - sp def and speed (correlation: -0.5) 

 **Suggestions** 
 1. For balance purposes, you could consider assigning a higher weight to taller Pokemon and a higher percentage of the total weight for Pokemon, as height and weight are strongly correlated with each other. 
 2. To ensure that the gameplay remains diverse and fair, you should also assign appropriate attack, defense, sp atk, sp def, and speed values to the Pokemon, taking into consideration their correlations with the other attributes. 
 3. Since the Pokemon attributes are strongly correlated, consider implementing a more complex system that takes advantage of this relationship, such as a network or genetic algorithm-based approach to balance the attributes of diverse Pokemon species. | **Findings** 
 1. The majority of Pokemon are Water (38.13%), followed by Normal (24.72%) and Grass (13.82%). 
 2. Fire, Psychic, and Steel types are relatively less frequent in the dataset. 
 3. Types with high positive correlations with main stats (HP, attack, defense, and speed) include Water, Normal, and Grass types, while types with high negative correlations include Fire, Psychic, and Steel types. 

 **Suggestions** 
 1. Consider balancing the game around the most common types (Water, Normal, and Grass) to make the gameplay more diverse and interesting for different types of strategies. 
 2. Since types with high positive correlations have higher HPs, attacks, and speeds, it may be beneficial to focus on the Pokemon with those stats to balance the game accordingly. 
 3. Monitor the performance of high-positive correlation types (Water, Normal, and Grass) to ensure their representation in the game doesn't lead to an imbalance in terms of stats or gameplay strategy. 
 4. Monitor the performance of high-negative correlation types (Fire, Psychic, and Steel) to ensure their representation in the game isn't detrimental due to a potential imbalance in game mechanics. |

Figure 5: **Case study.**

I will examine the relationships from the correlation matrix, including the most attended concerts and the conductor's experience, as well as the orchestras with the highest official ratings and their conductors.

```python
# Get the top 3 most attended concerts
top_attended = merged_df.nlargest(3, "Attendance")

# Get conductor names and age with most attended concerts
top_conductors = top_attended[["Name", "Age", "Year_of_Work", "Attendance"]].sort_values("Attendance", ascending=False)

print("Top 3 Conductors with Most Attended Concerts:")
print(top_conductors)
print()

# Get top 3 orchestras with the highest official ratings
top_rated_orchestras = merged_df.nlargest(3, "Official_ratings_(millions)")

# Get conductor names and orchestras with the highest official ratings
top_rated_conductors = top_rated_orchestras[["Orchestra", "Name", "Age", "Year_of_Work", "Official_ratings_(millions)"]].sort_values("Official_ratings_(millions)", ascending=False)

print("Top 3 Orchestras with Highest Official Ratings and Their Conductors:")
print(top_rated_conductors)
print()
```

Contribution RM: 5.05          Regularization RM: 5.15          Two RMs combined: 5.10

(a) A good case that receives high scores from both contribution RM and regularization RM.

Next, I will join the tables 'Order_Deliveries', 'Actual_Orders', 'Products', and 'Customers' to create a single DataFrame with the necessary information to analyze the most popular products across delivery routes.

```python
# Join tables
orders_deliveries = db["Order_Deliveries"]
actual_orders = db["Actual_Orders"]
products = db["Products"]
customers = db["Customers"]

# Merge tables
merged_df = pd.merge(orders_deliveries, actual_orders, on="actual_order_id")
merged_df = pd.merge(merged_df, products, left_on="actual_order_id", right_on="product_id")
merged_df = pd.merge(merged_df, customers, left_on="driver_employee_id", right_on="customer_id")

# Display a sample of 5 rows
print(merged_df.head())
```

Contribution RM: -2.63          Regularization RM: 6.59          Two RMs combined: 1.98

(b) A bad case that receives low score from contribution RM and high score from regularization RM.

Based on the analysis, provide a conclusion summarizing the findings and suggesting possible reasons for the high resettlement rates in the top 5 countries.

```python
print("Conclusion:")
print("The average resettlement rate in the top 5 countries is 593.8%, which is quite high compared to the average resettlement rate of 553.8% in the entire dataset.")
print("This is due to the high resettlement rates in the top 5 countries, which may be due to the following reasons:")
print("1. The top 5 countries have a higher population, which may indicate a higher demand for migration.")
print("2. The top 5 countries have specialized in Industry or Commerce, which might require a high number of workers to be migrated.")
print("3. The top 5 countries have a significant history of migration, which may indicate a diverse population and a need for migration.")
```

Contribution RM: 8.35          Regularization RM: -5.87          Two RMs combined: 1.24

(c) A reward hacking case that receives high score from contribution RM and low score from regularization RM.

Figure 6: **Qualitative examples** of code generations, and their scores assigned by reward models.

I have a database of [database title]. I am a stakeholder and I am analyzing the database to make a decision. Who am I and what decision might it be? List 10 possibilities in a numbered list.

Each point should introduce who I am and briefly explain my intention in this format: As a/the [who I am], I want to [explain my intention]

Examples:

Based on the extracurricular activities database:
1. As the dean of student affairs, I want to decide on extracurricular activities to promote or cut
2. As the department head, I want to decide on faculty advisor assignments
3. As the school administrator, I want to review and revise faculty activity engagement

Based on a diabetes database:
1. As a healthcare policy maker, I want to decide on healthcare resource allocation
2. As a NIH official, I want to decide on medical research funding
3. As a health insurance actuary, I want to improve health insurance pricing strategy
4. As a health provider, I want to decide on patient care and treatment

Based on an allergy database:
1. As a catering manager, I want to plan meal options
2. As the school principal, I want to plan allergy awareness programs
3. As an administrator in the Student Affairs or Housing department, I want to decide on housing assignments
4. As the school administrator, I want to improve campus emergency preparedness
5. As the school principal, I want to develop policies for allergy accommodations

Based on a Home Equity Line of Credit (HELOC) product database, you can:
1. As the credit risk manager, I want to modify the credit underwriting policy

The database is as follows:

Database `[title]` has [x] tables. Table names are: [aaa], [bbb], [ccc]

Table `[caption]` has [x] rows and [y] columns. Column are:
`[column name]`, example values: [value 1], [value 2], [value 3], [value 4], [value 5]
...

Table 9: Prompt for **query collection**.

I have a database of [database title]. As a [stakeholder role], I want to [describe intention].

Given below two findings/conclusions, which one is more helpful to my analysis?
* [answer bullet point 1]
* [answer bullet point 2]

Your response should be in the following format:
* Reasoning: <explain your reasoning here>
* Answer: <repeat the more helpful finding here>

Table 10: Prompt for **helpfulness annotation collection**.

I have a database of [database title]. As a [stakeholder role], I want to [describe intention].

I have hired two data analysts to perform the analysis, and they gave me two different reports (listed below). Each report consists of two lists, one for findings and one for suggestions. Which one is more helpful to my analysis? When evaluating helpfulness, you should consider the following three rubrics in decreasing priority: (1) relevance to my analysis goal; (2) insightfulness; and (3) diversity of perspectives, especially for suggestions.

Your response should be in the following format. Note: <answer> should be either Report-1 or Report-2
* Answer: <answer>
* Reasoning: <explain your reasoning here>

The reports are as follows:

# Report-1

[report 1]

# Report-2

[report 2]

Table 11: Prompt for **helpfulness evaluation**.

I have a database of [database title]. As a [stakeholder role], I want to [describe intention]. Below are the intermediate steps of their analysis. Which analysis is more informative? The more informative analysis should produce execution results that stick relevant to my analysis goal and bring more insights to my analysis.

Your response should be in the following format. Note: <answer> should be either Analysis-1 or Analysis-2
* Answer: <answer>
* Reasoning: <explain your reasoning here>

The reports are as follows:

# Analysis-1

[analysis 1]

# Analysis-2

[analysis 2]

Table 12: Prompt for **informativeness evaluation**.

