# OpenReview forum: "DACO: Towards Application-Driven and Comprehensive Data Analysis via Code Generation"
_ICML.cc/2024/Workshop/AI4MATH — ICML 2024 Workshop AI4MATH Poster_

### Official Review · Reviewer_EqMx · 2024-06-10

**Rating:** 6
**Confidence:** 3

**Summary:**

This paper focuses on the challenge of data analysis for tabular data, proposing new resources and benchmarks. Due to the high cost of expert-curated annotations, the authors use LLMs and a multi-turn prompting technique to automatically generate high-quality annotations. They introduce the DACO dataset, which includes 440 real-world databases, around 2,000 query-answer pairs, and a high-quality test set with human-refined annotations. A 6B parameter supervised fine-tuning (SFT) model trained on this dataset demonstrated reasonable data analysis capabilities. Further enhancement using reinforcement learning (DACO-RL algorithm) produced more helpful answers than the SFT model in 57.72% of cases, as validated by human annotators.

**Questions:**

In Section 2, under the Evaluation part, you mention that "The winning rate of each system is reported as a helpfulness score. To obtain a comparable set of numbers for all models, we report the winning rate of each model against TestA and TestH annotations. The upper bound for this score would be 50, as a score of 50 indicates that the model generations are perceived as helpful as annotations." I may not thoroughly understand this. If the upper bound for this score is 50, then why are there many values higher than 50 in Table 3 and Table 4? This seems a bit confusing. Could you please clarify?

**Reasons To Accept:**

1. The problem addressed by this paper is clearly defined and practical. The contributions appear to be extensive and valid.
2. The experiments, including ablation studies, are diverse and well-structured.
3. The demonstrations are clear and intuitive. For example, Table 2 (the distribution of DACO queries) effectively shows the diversity by analyzing verbs and noun objectives in the queries, which is reasonable.

**Reasons To Reject:**

1. The selection of the fine-tuned CodeGeeX-6B model for testing the algorithm seems unrepresentative, given the availability of many other high-quality code models of similar scale.
2. The paper explains the selection of 157 databases to maintain diversity, but it does not clarify the criteria for manually selecting 314 clean and interpretable databases from the initial 5,830. What criteria were used for selecting the remaining databases?
3. The similarity measure Sim(a,oi) between the final answer and code outputs is not persuasive for measuring helpfulness. Helpfulness should differ from "similarity," and a more appropriate metric should be proposed. Additionally, the paper introduces "informativeness" as another metric for evaluating code generation quality in the ablation study. What is the relationship between these two metrics? Is there a way to combine them, and what is the rationale for defining a separate metric?

---

### Official Review · Reviewer_qwE6 · 2024-06-11

**Rating:** 5
**Confidence:** 5

**Summary:**

- The study's contributions include constructing the DACO dataset using a multi-turn prompting technique on diverse real-world databases, curating a human-refined evaluation set for benchmarking models, and designing the DACO-RL algorithm to optimize code and answer generation towards human alignment, achieving a 57.72% win rate in human evaluations of helpfulness.

**Questions:**

see above

**Reasons To Accept:**

- This study proposes three reward models to optimize language model performance: Ra, Rc, and Rr. Ra evaluates the helpfulness of individual bullet points in the final answer through pairwise comparisons. Rc assesses the helpfulness of intermediate steps based on their similarity to the final answer. Rr penalizes reward hacking behaviors to ensure genuine helpfulness. The combined rewards guide the Proximal Policy Optimization (PPO) algorithm, enhancing both intermediate steps and final answers in multi-turn conversational settings by maximizing generalized advantage estimation (GAE).

- DACO-RL achieves a 57.72% win rate over SFT in human evaluations.

- DACO-RL better addresses user queries, providing more relevant and insightful responses, as illustrated in Figure 5.

**Reasons To Reject:**

- Helpfulness is defined as similarity; however, similarity only captures how closely the outputs resemble each other, not their practical utility or relevance to the user's query.

- The DACO-RL method is tested on fine-tuned CodeGeeX-6B model. Other state of the art code generation models could be compared.

- Other datasets could be evaluated.

- The pipeline proposed could use further clarification: "To improve the quality of such automatically constructed annotations, we additionally allow GPT-4 to correct its own mistakes when its generated code leads to run-time or syntax error, where only the corrected codes are kept." It is unclear how the GPT corrects it mistakes?; "At each turn, the LLM will produce a python code snippet and take its execution outputs as evidences to reason over and support its follow-up interpretation. After each turn, we prompt the model to decide whether the analysis is sufficiently comprehensive; if deemed sufficient, it terminates the coding turns and produces the final answer." This needs further clarification, e.g. what is sufficient?;  Throughout the paper, is is unclear what is bullet point refering to?

- "execute the code with a code interpreter" typo --> executes

- The study could consider different presentation using using w/ as opposed to w/o in Table 4 with rewards.

- The proposed method, DACO-RL, is evaluated on two dataset InfiAgent-DA benchmark, and the FeTaQA dataset perfoming well on one over other. Is is unclear why the algorithm was not performing well for information lookup tasks.

- The study finds that Rc and Rr encourage more diverse and helpful code generation, and Rc favors APIs that extract important information from data, such as nlargest, sort_values, mean, and print. It is unclear how these rewards were created and the need based on the related work.

- The dataset can be shared to encourage reproducibility, and metrics/rewards to validate results.

---

### Meta-Review · Area_Chair_ibkd · 2024-06-13

**Recommendation:** Accept (Poster)
**Confidence:** 4

**Metareview:**

The paper constructs a new benchmark for data analysis in real-world databases. It also proposes a reinforcement learning based method that can better address user queries. The author could further improve the paper by reformulating helpfulness, testing with more high-quality code generation models, and clarifying the pipeline.

---

### Decision · Program_Chairs · 2024-06-13

Accept (Poster)